# Mice lacking triglyceride synthesis enzymes in adipose tissue are resistant to diet-induced obesity

Chandramohan Chitraju[1,2], Alexander W Fischer[1,2], Yohannes A Ambaw[1,2,3], Kun Wang[1,2], Bo Yuan[1], Sheng Hui[1], Tobias C Walther[1,2,3,4,5]*†, Robert V Farese Jr[1,2,3,4]*†

[1]Department of Molecular Metabolism, Harvard T.H. Chan School of Public Health, Boston, United States; [2]Department of Cell Biology, Harvard Medical School, Boston, United States; [3]Cell Biology Program, Sloan Kettering Institute, Memorial Sloan Kettering Cancer Center, New York, United States; [4]Broad Institute of Harvard and MIT, Cambridge, United States; [5]Howard Hughes Medical Institute, Boston, United States

*For correspondence:
twalther@mskcc.org (TCW);
rfarese@mskcc.org (RVF)

†These authors contributed equally to this work

Competing interest: The authors declare that no competing interests exist.

**Abstract** Triglycerides (TGs) in adipocytes provide the major stores of metabolic energy in the body. Optimal amounts of TG stores are desirable as insufficient capacity to store TG, as in lipodystrophy, or exceeding the capacity for storage, as in obesity, results in metabolic disease. We hypothesized that mice lacking TG storage in adipocytes would result in excess TG storage in cell types other than adipocytes and severe lipotoxicity accompanied by metabolic disease. To test this hypothesis, we selectively deleted both TG synthesis enzymes, DGAT1 and DGAT2, in adipocytes (ADGAT DKO mice). As expected with depleted energy stores, ADGAT DKO mice did not tolerate fasting well and, with prolonged fasting, entered torpor. However, ADGAT DKO mice were unexpectedly otherwise metabolically healthy and did not accumulate TGs ectopically or develop associated metabolic perturbations, even when fed a high-fat diet. The favorable metabolic phenotype resulted from activation of energy expenditure, in part via BAT (brown adipose tissue) activation and beiging of white adipose tissue. Thus, the ADGAT DKO mice provide a fascinating new model to study the coupling of metabolic energy storage to energy expenditure.

## eLife assessment

This study introduces a **valuable** paradigm in the field of adipose tissue biology: blocking triglyceride storage in adipose tissue does not lead to lipodystrophy and impaired glucose homeostasis but instead improves metabolic health. The evidence supporting these claims is **convincing**, based on a comprehensive metabolic analysis, although mechanistic studies would strengthen the study and its impact. This study will be of high interest to those in the adipose tissue biology and metabolism fields.

## Introduction

Because energy sources are not always available from the environment, many metazoan organisms have evolved the ability to store large amounts of metabolic energy as triglycerides (TGs) in adipose tissue. TG is particularly optimal for energy storage because it serves as stores of highly reduced carbon and does not require water for its storage. In cells, such as adipocytes, TGs are stored in organelles called lipid droplets (LDs). In adipocytes of mammals, TGs are stored in a unilocular adipocyte

that fills the majority of the aqueous cytoplasm. Although TGs can also be found in LDs in other cell types (i.e., myocytes, hepatocytes, enterocytes), adipocytes represent by far the major energy depots in mammals.

Abundant evidence from many studies suggests that there is an optimal range for adipocyte TG storage in an organism. Exceeding the capacity to store TG in adipocytes occurs in obesity and is often accompanied by deposition of TG in other tissues and metabolic diseases, such as diabetes mellitus or non-alcoholic fatty liver disease. Conversely, insufficient TG storage such as occurs in lipodystrophy is usually associated with adipocyte endocrine deficiency and similar metabolic derangements.

Here, we sought to test the requirement for TG storage in adipocytes in murine physiology at baseline and in response to high-fat feeding. We generated mice lacking both known TG synthesis enzymes, DGAT1 and DGAT2 (*Cases et al., 1998*; *Cases et al., 2001*), in adipocytes. We expected to generate a mouse model similar to those of classic lipodystrophy due to defects of TG storage in adipose tissue. Moreover, we hypothesized that these mice would have accumulation of TGs in other tissues, such as the liver or skeletal muscle, resulting in lipotoxicity and metabolic derangements, such as insulin resistance or fatty liver disease. To our surprise, we found the opposite result. We report here that selectively impairing TG storage in adipocytes leads to a unique murine model in which depletion of energy stores is not accompanied by metabolic derangements but instead results in protection from adverse metabolic effects, even with high-fat diet (HFD) feeding, due to activation of energy dissipation pathways.

## Results

### ADGAT DKO mice have reduced fat mass and TGs in adipose tissue

To generate mice lacking TGs in adipose tissue (ADGAT DKO), we crossed adipose tissue-specific *Dgat1* knockout mice (Cre-transgene expressed under control of the mouse adiponectin promoter [*Eguchi et al., 2011*] with *Dgat2* flox mice [*Chitraju et al., 2019*]). Validation of gene knockouts showed mRNA levels of *Dgat1* were decreased by ~95% and *Dgat2* by ~90% in both inguinal white adipose tissue (iWAT) and interscapular brown adipose tissue (BAT) (*Figure 1—figure supplement 1A*). Western blot analysis showed that DGAT1 and DGAT2 proteins were absent in iWAT and BAT (*Figure 1—figure supplement 1B*). In vitro DGAT activity in lysates of adipose tissue of ADGAT DKO mice was decreased by ~80% in iWAT and ~95% in BAT (*Figure 1—figure supplement 1C*). Similarly, in vitro DGAT activity in isolated adipocytes of iWAT was decreased by ~80% (*Figure 1—figure supplement 1D*).

ADGAT DKO mice appeared healthy (*Figure 1A*) and yielded offspring with the predicted Mendelian ratio of genotypes. Nuclear magnetic resonance imaging showed that fat depots were decreased in chow-fed ADGAT DKO mice (*Figure 1B*). Body weights of 12-week-old chow-fed control mice (*Dgat1* and *Dgat2* double-flexed mice, D1D2 flox) and ADGAT DKO mice were similar (*Figure 1C*), but dual-energy X-ray absorptiometry (DEXA) analysis revealed that the fat mass was decreased by ~60% in ADGAT DKO mice. The reduction in fat mass was persistent: DEXA analysis of 1-year-old ADGAT DKO mice showed that the fat mass was reduced by ~75% and that lean mass was increased by ~15% (*Figure 1—figure supplement 1E*). Visceral white adipose tissue (WAT) depots (gonadal, mesenteric, pericardial, and perirenal fat depots) were markedly atrophied in ADGAT DKO mice (*Figure 1D*, *Figure 1—figure supplement 1F*). Gonadal WAT (gWAT) and subcutaneous iWAT in ADGAT DKO mice appeared distinctly 'beige' in color (*Figure 1D*, *Figure 1—figure supplement 1F*). In ADGAT DKO mice, iWAT and BAT were denser, as demonstrated by their sinking in a liquid fixative (*Figure 1F*).

The interscapular BAT depot in ADGAT DKO mice appeared darker brown than that in control mice (*Figure 1—figure supplement 2A*). TGs and LDs were undetectable in BAT of ADGAT DKO mice (*Figure 1—figure supplement 2B and C*). Positron emission tomography-computed tomography scanning using $^{18}$-fluoro-deoxyglucose ($^{18}$-FDG-PET/CT) showed that, after injection of β-3-adrenoceptor agonist (CL 316,243), BAT of chow-fed ADGAT DKO mice took up more glucose than BAT of control mice (*Figure 1—figure supplement 2D*), presumably to fuel thermogenesis. In agreement with increased glucose uptake, glycogen levels in BAT of ADGAT DKO mice were increased in all conditions except 4°C cold exposure (*Figure 1—figure supplement 2E*). The latter condition may reflect increased glycogen requirements in BAT of ADGAT DKO mice to maintain thermogenesis. This

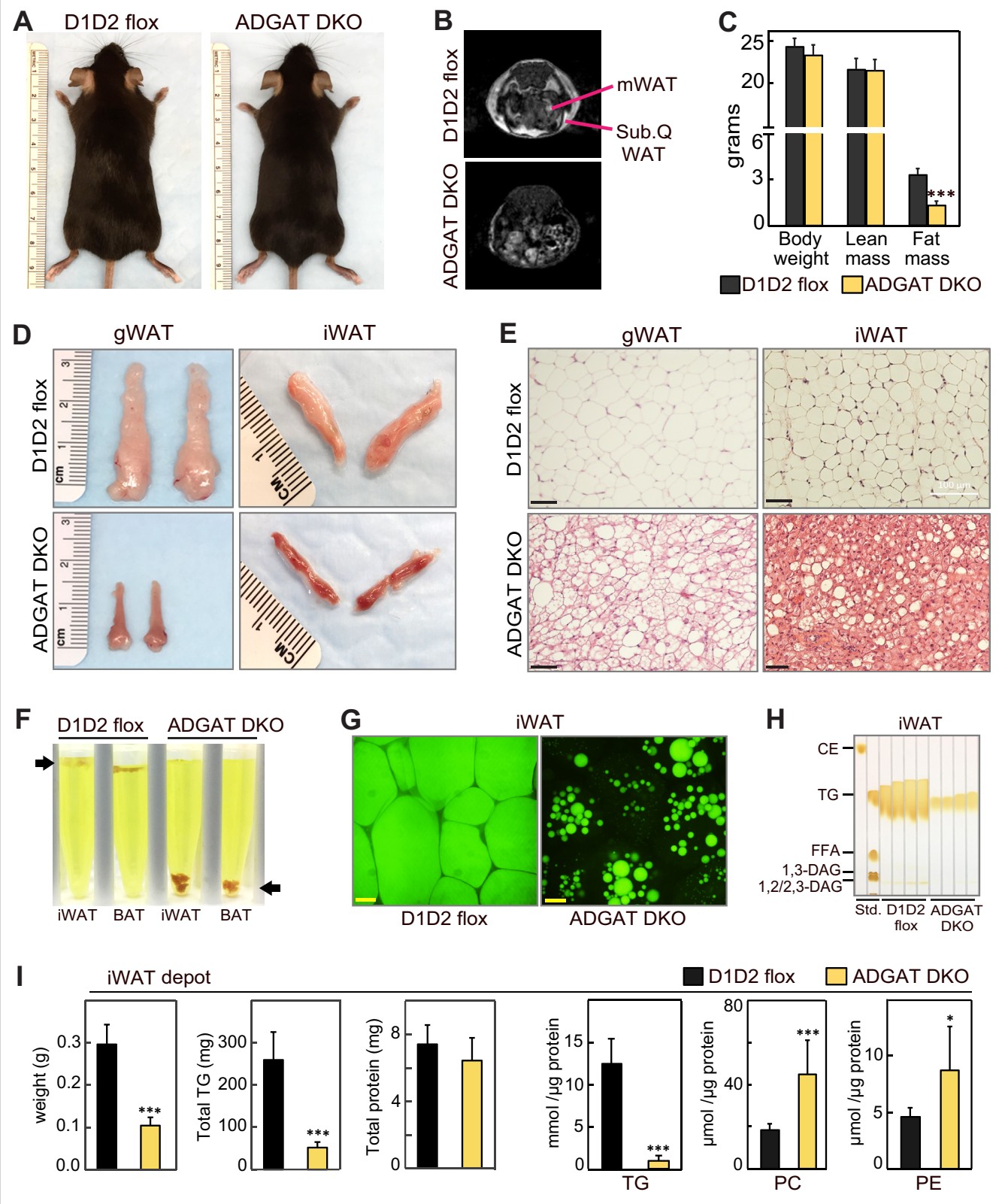

**Figure 1.** ADGAT DKO mice have reduced fat mass and triglycerides in adipose tissue. (**A**) Chow-diet-fed ADGAT DKO mice appear normal. Representative photographs of control and ADGAT DKO female mice fed a chow diet. (**B**) Fat depots were decreased in ADGAT DKO mice. Nuclear magnetic resonance imaging of chow-diet-fed male mice. mWAT, mesenteric white adipose tissue; Sub.Q WAT, subcutaneous white adipose tissue. (**C**) ADGAT DKO mice have decreased fat mass. Dual-energy X-ray absorptiometry (DEXA) analysis of lean mass and fat mass of chow-diet-fed male

*Figure 1 continued on next page*

*Figure 1 continued*

mice (n=8). (**D**) Fat depots were atrophied in ADGAT DKO mice. Representative photographs of inguinal white adipose tissue (iWAT) and gonadal white adipose tissue (gWAT) from male mice (n=8). (**E**) gWAT and iWAT of ADGAT DKO mice contain multi-locular lipid droplets in adipocytes. H&E-stained sections of gWAT and iWAT from chow-diet-fed male mice (n=6). Scale bars, 50 μm. (**F**) WAT and brown adipose tissue (BAT) of ADGAT DKO mice were denser than controls and sink in an aqueous buffer with fixative (1.25% formaldehyde, 2.5% glutaraldehyde, and 0.03% picric acid in 0.1 M sodium cacodylate buffer, pH 7.4, density = 1.01 g/ml) used to fix tissue for electron microscopy. (**G**) Lipid droplets (LDs) in iWAT of ADGAT DKO mice stain with BODIPY. Confocal fluorescence microscopy images of adipose tissue. LDs were stained by BODIPY 493/503. Scale bar, 25 μm. (**H**) WAT of ADGAT DKO mice contain triglycerides. Thin layer chromatography analysis of lipids from iWAT of male mice (n=4). TGs, triglycerides; CE, cholesterol esters, FFA, free fatty acids; DAG, diacylglycerol. (**I**) Increased phospholipid levels in iWAT of ADGAT DKO mice. Lipids in iWAT were extracted and quantified by mass spectrometry (n=8). Data are presented as mean ± SD. *p<0.05, ***p<0.001.

The online version of this article includes the following figure supplement(s) for figure 1:

**Figure supplement 1.** ADGAT DKO mice have reduced DGAT1 and DGAT2 expression in adipose tissue.

**Figure supplement 2.** ADGAT DKO mice have reduced triglycerides in brown adipose tissue (BAT).

**Figure supplement 3.** Most of the triglyceride molecular species were decreased in adipose tissue of ADGAT DKO mice.

phenotype of DGAT-deficient BAT exhibiting increased glucose uptake and glycogen stores as an alternate fuel is consistent with our previous findings of BAT-specific knockout of both DGAT enzymes (*Chitraju et al., 2020*).

## ADGAT DKO mice gradually activate an alternative mechanism to synthesize and store TGs

DGAT1 and DGAT2 appear to account for most of TG synthesis in mice. Newborn mice lacking both DGAT enzymes have >95% reduction in whole body TGs (*Stone et al., 2004*), and adipocytes derived from fibroblasts lacking both enzymes fail to accumulate TGs or LDs (*Harris et al., 2011*). In agreement with this, histological analysis of WAT of 8-week-old ADGAT DKO mice showed fewer and much smaller LDs than control WAT (*Figure 1E*). However, by age 15 weeks, ADGAT DKO mice exhibited more LDs in iWAT than 8-week-old ADGAT DKO mice, suggesting that they activate alternate pathways to accumulate neutral lipids (*Figure 1—figure supplement 3A*). This finding was more prominent in iWAT than in BAT. The neutral lipid that accumulated was BODIPY-positive (*Figure 1G*), and TLC analysis of adipose tissue lipids from 15-week-old ADGAT DKO mice revealed the lipids to be TGs (*Figure 1H*). Feeding ADGAT DKO mice an HFD increased levels of TGs by ~twofold in iWAT at age 15 weeks, but the TG content of iWAT remained ~70% less than control mice (*Figure 1—figure supplement 3A*, *Figure 4—figure supplement 1A*). The mass reduction of iWAT fat pads was accounted for predominantly by a decrease in TG mass per fat pad (*Figure 1I*); protein levels per fat pad were similar to controls, and no other neutral lipids were detected. Lipid analyses of iWAT by mass spectrometry revealed that TG levels were reduced by ~80% across all detected TG species (*Figure 1I*, *Figure 1—figure supplement 3C*). In contrast, several phospholipids were substantially increased in iWAT of ADGAT DKO mice (*Figure 1I*), which may have contributed to the residual fat mass of the iWAT fat pads. Enzyme assays revealed that adipose tissues and isolated white adipocytes from 15-week-old chow-diet-fed ADGAT DKO mice had detectable (~20% of normal) DGAT activity in iWAT that was not inhibited by DGAT1- or DGAT2-specific inhibitors (*Figure 1—figure supplement 1C and D*). These data suggest that deletion of DGAT1 and DGAT2 in WAT induces a DGAT activity from an alternative enzyme, possibly from other candidates in the DGAT2 gene family (*Yen et al., 2005*). mRNA levels for MGAT1 and MGAT2, enzymes in the same protein family as DGAT2, were increased in iWAT of ADGAT DKO mice and thus these proteins are candidates for this activity (*Figure 1—figure supplement 1F and G*).

## Adipose tissue TG stores are required to maintain activity and body temperature during fasting

Because ADGAT DKO mice have severely decreased TG stores, we expected that they would not tolerate fasting well. After 14 hr of fasting, 15-week-old ADGAT DKO mice had lost 10% of body weight (vs. control mice) (*Figure 2A*) and entered a torpor-like state with decreased physical activity and huddling together (*Video 1*). Fasting of ADGAT DKO mice also resulted in hypothermia, with body temperatures dropping to ~30°C (*Figure 2B*), a phenotype exacerbated by cold exposure (*Figure 2E*

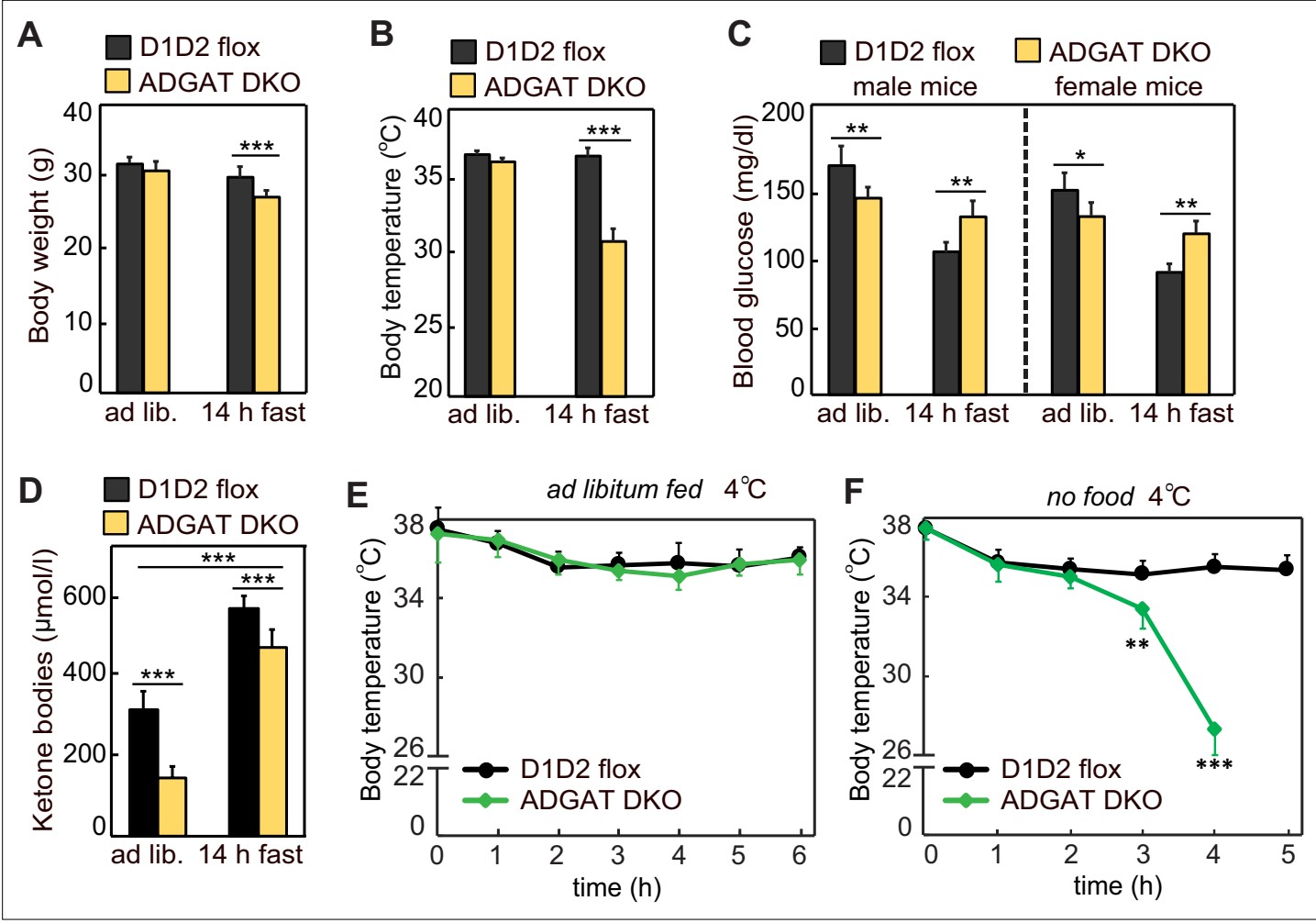

**Figure 2.** Adipose tissue triglyceride (TG) stores are required to prevent a torpor-like state during fasting. (**A**) Body weights of male mice fed ad libitum or fasted 14 hr (n=10). Ad lib., ad libitum fed. (**B**) Core body temperature of male mice housed at room temperature and fed ad libitum or fasted for 14 hr (n=10). (**C**) Blood glucose levels in male and female mice fed ad libitum or fasted for 14 hr (n=8). (**D**) Levels of plasma ketone bodies in male mice fed ad libitum or fasted for 14 hr (n=8). (**E and F**) Core body temperature of male mice housed in cold (5°C) with or without food (n=10). Data are presented as mean ± SD. *p<0.05, **p<0.01, ***p<0.001.

*and F*). This differs from the phenotype of BAT DGAT DKO mice, which maintain body temperature during fasting (*Chitraju et al., 2020*), presumably because energy stores in WAT are present. Fasting levels of ketone bodies were ~10% lower, and glucose levels were moderately higher in ADGAT DKO mice than control mice (*Figure 2C and D*), possibly reflecting their greater dependency on glucose as fuel. Thus, as expected, deletion of TG stores in adipose tissue resulted in reduced fuel stores that dramatically altered the physiological responses of the mice to fasting or cold.

## Endocrine function of WAT is maintained in ADGAT DKO mice

Lipodystrophy is a metabolic disease characterized by altered fat distribution, often with severely reduced amounts of adipose tissue and TG storage. Classically, lipodystrophy in humans and mice is accompanied by reduced levels of adipocyte-derived endocrine hormones and often results in insulin resistance and diabetes (*Oral et al., 2002*; *Reue and Phan, 2006*; *Péterfy et al., 2001*). ADGAT DKO mice share an impaired capacity to store TG in adipose tissue with lipodystrophy models. However, despite this, in these mice adiponectin and leptin mRNA levels were moderately increased in iWAT (*Figure 3A*), whereas the mRNA levels of *Plin1* were unchanged (*Figure 3A*), and plasma levels of adiponectin and leptin were normal and 40% decreased, respectively (*Figure 3B*). When normalized to adipose tissue weight, leptin levels were similar to control mice (*Figure 3C*). Glucose levels in

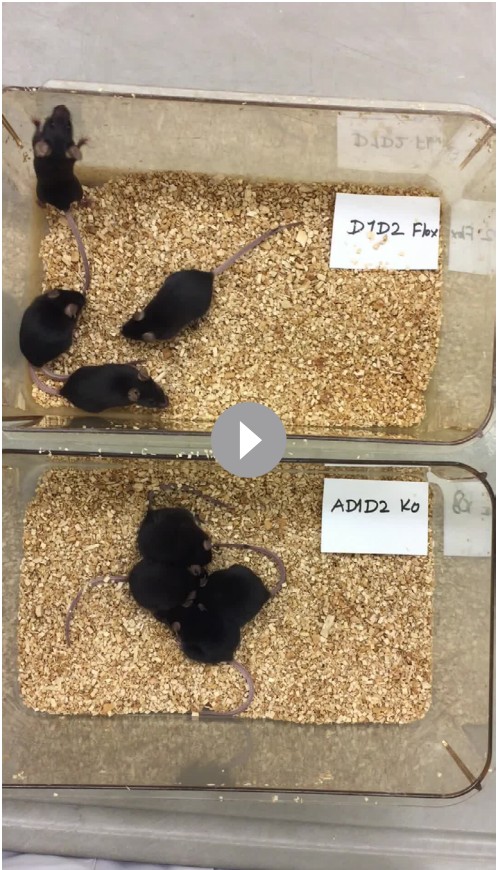

**Video 1.** ADGAT DKO mice enter torpor during fasting. After 10 hr of fasting ADGAT DKO mice enter a torpor-like state with decreased physical activity and huddling together. Fasting of ADGAT DKO mice also resulted in hypothermia, with body temperatures dropping to ~30°C.

https://elifesciences.org/articles/88049/figures#video1

ADGAT DKO mice fasted for 4 hr were slightly lower (169±16 mg/dl vs. 144±14 mg/dl, respectively, p<0.01) than in control mice, and insulin levels were not different (*Figure 3D*). Analysis of plasma metabolites showed an ~30% reduction in non-esterified fatty acids, an ~15% reduction in glycerol (*Figure 3E*), and an ~50% reduction in ketones in chow-diet-fed ADGAT DKO mice (*Figure 3F*). Glucose and insulin tolerances were similar in chow-diet-fed ADGAT DKO and control mice (*Figure 3G and H*). Thus, despite the impairment of fat storage in adipose tissue, ADGAT DKO mice had substantial levels of adipocyte-derived endocrine hormones and apparently normal glucose metabolism.

Lipodystrophy is also typically accompanied by ectopic lipid deposition, particularly manifesting as hepatic steatosis (*Cui et al., 2011*; *Moitra et al., 1998*; *Shimomura et al., 1998*). In contrast, livers of ADGAT DKO mice appeared normal (*Figure 3I*), with moderately increased weights in 15-week-old chow-diet-fed mice (1.7±0.2 g vs. 2.1±0.3 g, p<0.05). TG levels were only modestly increased (~10%) in livers of ADGAT DKO mice and were unchanged in skeletal muscle (*Figure 3J*). Thus, ADGAT DKO mice, with markedly reduced TG storage in adipocytes, were remarkably metabolically healthy, with essentially none of the metabolic derangements typically associated with lipodystrophy.

## ADGAT DKO mice are resistant to diet-induced obesity and associated metabolic derangements

We next tested whether the metabolically healthy phenotype of ADGAT DKO mice would persist with feeding of a western-type HFD, which normally causes obesity and insulin resistance. We hypothesized that fatty acids from the HFD would not be stored in adipocytes of ADGAT DKO mice and as a consequence ectopically accumulate, resulting in tissue lipotoxicity. However, after feeding ADGAT DKO mice an HFD for 12 weeks, they appeared healthy and remained relatively lean, with both male and female mice gaining ~40% less body weight than control D1D2 flox mice (*Figure 4A–C*). The reduction in body weight was due to an ~70% reduction in fat mass (*Figure 4C*). Food intake during HFD feeding was similar (*Figure 4D*), implying ADGAT DKO mice have increased energy expenditure. This was validated by indirect calorimetry, where ADGAT DKO mice exhibited increased energy expenditure that was particularly prominent during night-time, when the mice were eating (*Figure 4E*). The respiratory exchange ratio was lower in ADGAT DKO mice during HFD feeding (*Figure 4F*), consistent with increased fat oxidation. We did not measure caloric loss in the feces of ADGAT DKO mice and would not expect this with adipocyte-specific deletions of DGAT1 and DGAT2.

We also examined metabolic parameters in the HFD-fed ADGAT DKO mice. Plasma glucose levels were slightly lower in ADGAT DKO, and insulin levels were not different (*Figure 4—figure supplement 1B and C*). ADGAT DKO mice were protected from HFD-induced glucose intolerance (*Figure 4G*). The insulin response was similar in ADGAT DKO and control mice, although the ADGAT DKO mice basal levels of glucose were reduced (*Figure 4H*). Liver weights and hepatic TG levels were markedly increased with HFD in both ADGAT DKO mice and controls and were ~20% and ~10% higher

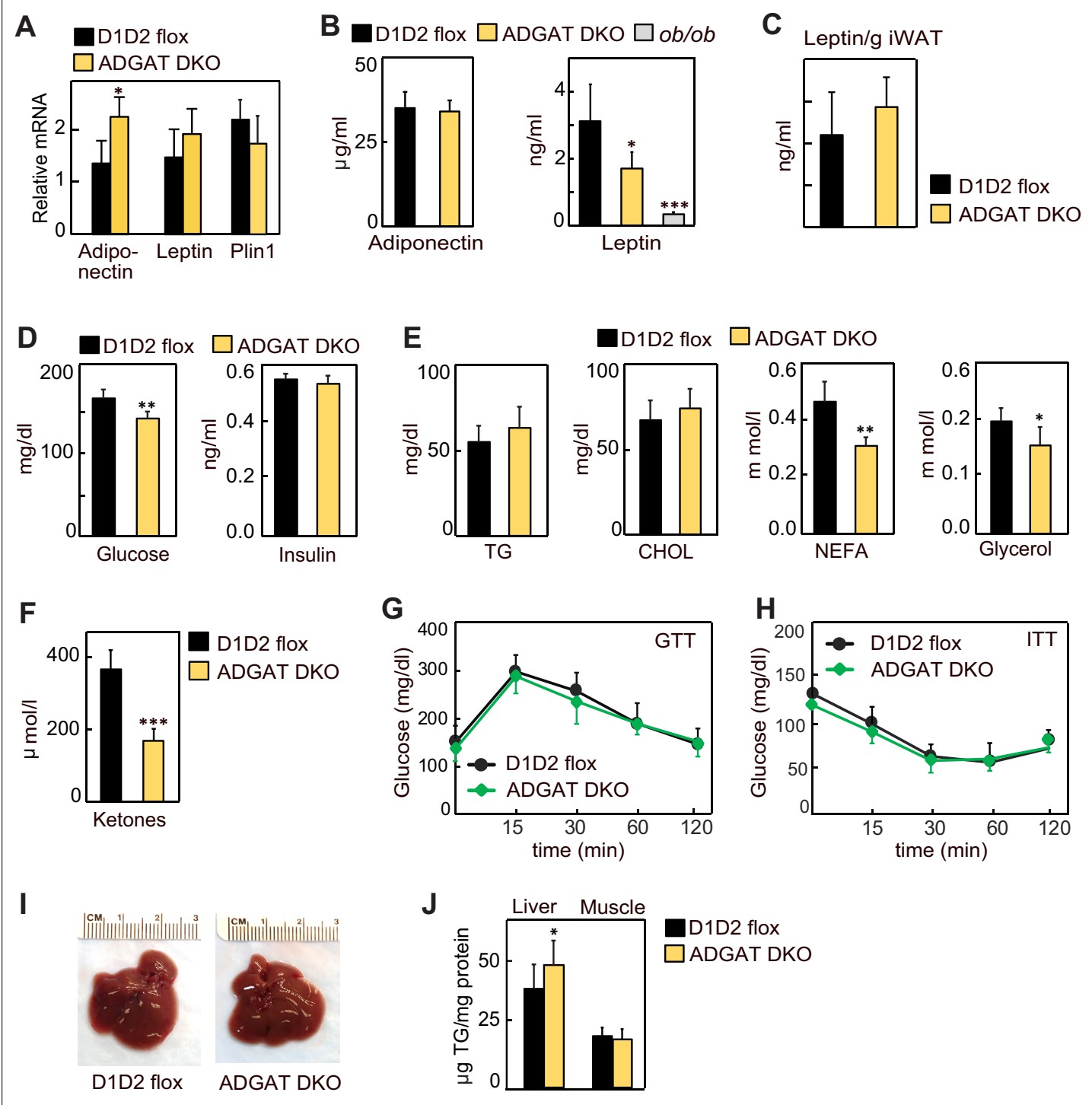

**Figure 3.** Lipodystrophy is uncoupled from detrimental metabolic effects in ADGAT DKO mice. (**A**) Adiponectin and leptin mRNA levels were moderately increased in inguinal white adipose tissue (iWAT) of ADGAT DKO mice. Relative mRNA levels of leptin and adiponectin in iWAT of chow-diet-fed male mice (n=6). (**B**) Plasma levels of adiponectin were normal, and leptin levels were moderately decreased in ad libitum chow-diet-fed male mice (n=8). (**C**). Plasma leptin levels normalized per gram of WAT mass (n=8). (**D**) ADGAT DKO mice had normal glucose and insulin levels. Glucose and insulin levels in ad libitum chow-diet-fed male mice (n=8). (**E**) Decreased free fatty acids in ADGAT DKO mice. Levels of plasma metabolites in ad libitum chow-diet-fed male mice (n=8). (**F**) Decreased ketones in ad libitum chow-diet-fed ADGAT DKO male mice (n=8). TG, triglycerides; CHOL, cholesterol; NEFA, non esterified fatty acids (**G and H**) Glucose and insulin tolerance tests were normal in chow-diet-fed male mice (n=10). (**I**) ADGAT DKO mice had non-steatotic livers. Representative photographs of livers from chow-diet-fed mice. (**J**) Triglycerides were moderately increased in livers of ADGAT DKO mice. Triglyceride levels in livers and skeletal muscle of chow-diet-fed male mice (n=6). Data are presented as mean ± SD. *p<0.05, **p<0.01, ***p<0.001.

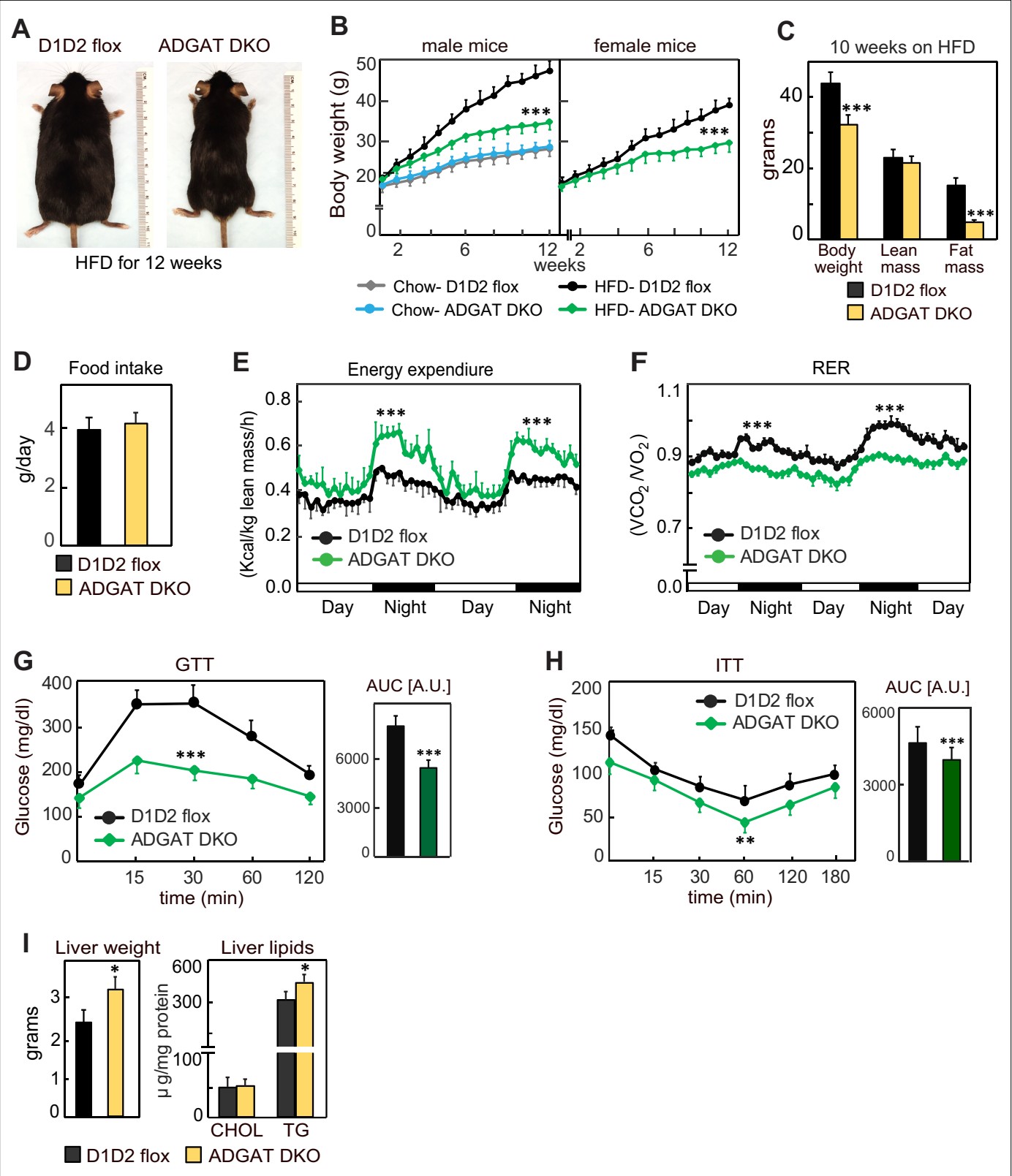

**Figure 4.** ADGAT DKO mice are resistant to diet-induced obesity and glucose intolerance. (**A**) ADGAT DKO mice stay lean on a high-fat diet (HFD). Representative photographs of male mice fed on HFD for 12 weeks. (**B**) Both male and female ADGAT DKO mice gained ~40% less body weight than control mice. Body weights of mice fed on a chow-diet or HFD (n=15 for males, n=12 for females). (**C**) ADGAT DKO mice had decreased fat mass on HFD feeding. Dual-energy X-ray absorptiometry (DEXA) analysis of lean mass and fat mass of HFD fed male mice (n=10). (**D**) ADGAT DKO male

*Figure 4 continued on next page*

Figure 4 continued

mice had normal food intake during HFD feeding (n=5). (**E and F**) ADGAT DKO mice had increased energy expenditures. Energy expenditure and respiratory quotient on HFD-fed male mice measured by indirect calorimetry. Values were normalized to lean mass (n=4). (**G and H**) ADGAT DKO mice were protected from HFD-induced glucose intolerance and insulin resistance. Glucose- and insulin-tolerance tests were performed on HFD-fed (for 9 or 10 weeks, respectively) male mice (n=10). AUC, area under the curve. (**I**) Liver weights and triglyceride levels were moderately increased in HFD-fed ADGAT DKO male mice (n=6). Data are presented as mean ± SD. *p<0.05, **p<0.01, ***p<0.001.

The online version of this article includes the following figure supplement(s) for figure 4:

**Figure supplement 1.** ADGAT DKO mice gradually activate an alternative mechanism to synthesize triglycerides.

in ADGAT DKO mice, respectively (*Figure 4I*). Hepatic cholesterol levels were similar (*Figure 4I*). HFD-induced activation of ER-stress response in the livers was similar to control mice (*Figure 4—figure supplement 1D*). Thus, surprisingly, despite not being able to robustly store TGs in adipocytes, ADGAT DKO mice were resistant to most effects of an HFD, and our studies indicate that they activate compensatory mechanisms of energy expenditure that increase fat oxidation.

## ADGAT DKO mice activate energy dissipation mechanisms, including adipocyte beiging in WAT

We next investigated the mechanisms for improved metabolic health in ADGAT DKO mice. Browning or beiging of adipose tissue is associated with improved metabolic health in mice and humans (*Becher et al., 2021*; *Wu et al., 2012*; *van Marken Lichtenbelt et al., 2009*; *Kazak et al., 2015*), and the 'beige' appearance in iWAT and gWAT depots of ADGAT DKO mice (*Figure 1D and E*) suggested that beiging adaptations may be present. Histological examination showed almost all adipocytes in both iWAT and gWAT contained multi-locular LDs (*Figure 5A and C*). mRNA levels of signature genes of adipocyte beiging, such as *Ucp1* (~600-fold), *Idea* (~20-fold), *Ppara* (~10-fold), and *Pgc1a* (~6-fold), were markedly increased in iWAT and gWAT of room temperature housed chow-fed ADGAT DKO mice (*Figure 5B and D*). Fatty acid levels were decreased; intermediates of glycolysis and Krebs cycle were enriched in both iWAT and BAT of ADGAT DKO mice, consistent with increased glycolysis and fatty acid oxidation (*Figure 5—figure supplement 1A and B*). Protein levels of UCP1 and respiratory complex proteins were also markedly greater in iWAT of room-temperature-housed chow-fed ADGAT DKO mice than controls and were even further increased under HFD conditions (*Figure 5E*). Beiging of adipocytes in iWAT appeared independent of ambient temperature and was also present in ADGAT DKO mice after 6 weeks of thermoneutral housing (*Figure 5F*), and blood glucose levels were moderately lower in thermoneutral-housed male and female mice (*Figure 5G*). Beiging appeared to be non-cell-autonomous, as the changes found in beige fat were largely absent in cells differentiated from pre-adipocytes, with the exception of a twofold increase of *Ucp1* mRNA levels (*Figure 5—figure supplement 2A–D*), suggesting that beiging in ADGAT DKO mice is activated in part through the sympathetic nervous system (SNS) *Kajimura et al., 2015*; *Harms and Seale, 2013*. Hormones, such as FGF21, also can activate beiging, either via the SNS (*Fisher et al., 2012*; *Owen et al., 2014*) or in a paracrine manner (*Fisher et al., 2012*; *Abu-Odeh et al., 2021*). FGF21 mRNA levels were increased by ~twofold and ~sixfold in liver and iWAT of ADGAT DKO mice, respectively (*Figure 5—figure supplement 3A*), and plasma levels of FGF21 were increased ~threefold in ADGAT KO mice (*Figure 5—figure supplement 3B*). However, plasma FGF21 levels were similar in ADGAT DKO mice and controls that were fed an HFD, suggesting an endocrine FGF21 effect is not responsible for the increased beiging of iWAT (*Figure 5—figure supplement 3B–D*).

## Discussion

We report a novel mouse model with impaired TG synthesis in adipocytes. The resultant defect in TG stores had both expected and surprising effects on murine physiology. First, as expected, ADGAT DKO mice did not tolerate fasting well. At room temperature, fasted ADGAT DKO mice entered a torpor-like state, characterized by decreased ambulation and a drop in body temperature. Torpor is a physiological state that enables conservation of metabolic energy and the signals to induce this state are poorly understood (*Gavrilova et al., 1999*). The phenotype of ADGAT DKO mice suggests that depletion of adipocyte TG stores is sufficient to induce energy conservation and is somehow sensed.

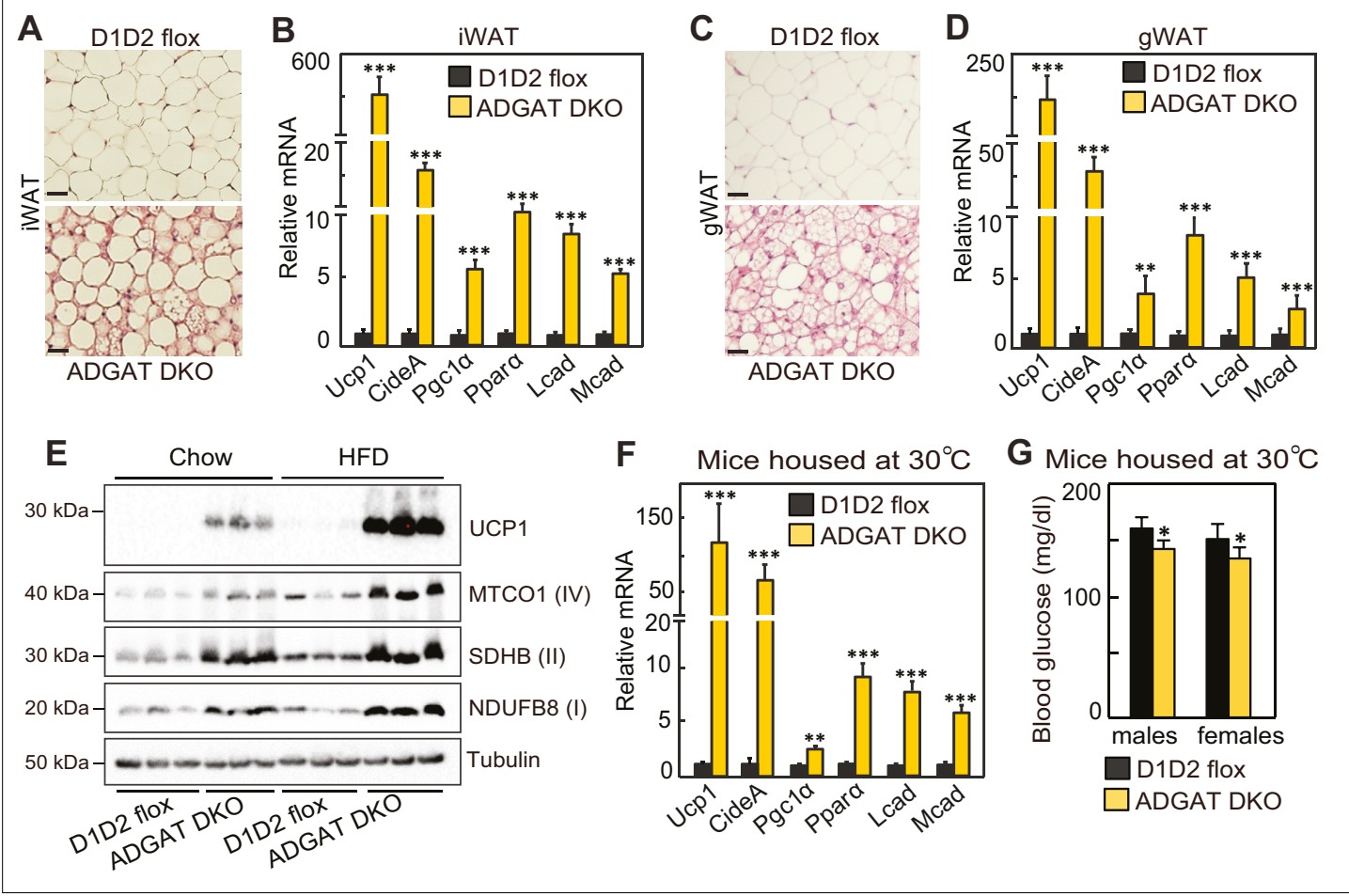

**Figure 5.** Beiging of white adipose tissue (WAT) in ADGAT DKO mice. (**A**) ADGAT DKO mice contain multi-locular lipid droplets (LDs) in adipocytes of inguinal WAT (iWAT). H&E-stained sections of iWAT from male mice fed a chow diet and housed at room temperature (n=6). Scale bars, 50 μm. (**B**) Increased expression of thermogenic marker genes in iWAT of ADGAT DKO mice. Relative mRNA levels of thermogenic genes in iWAT of male mice fed a chow diet and housed at room temperature (n=6). (**C**) ADGAT DKO mice contain multi-locular LDs in adipocytes of gonadal white adipose tissue (gWAT). H&E-stained sections of gWAT from male mice fed a chow diet and housed at room temperature (n=6). Scale bars, 50 μm. (**D**) Increased expression of thermogenic marker genes in gWAT of ADGAT DKO mice. Relative mRNA levels of thermogenic genes in gWAT of male mice fed a chow diet and housed at room temperature (n=6). (**E**) High-fat diet (HFD) feeding increases levels of UCP1 in iWAT of ADGAT DKO mice. Immunoblot analysis of UCP1 and OXPHOS proteins in iWAT of male mice fed either a chow diet or an HFD (n=3). Mice were housed at room temperature. (**F**) Beiging was intact in iWAT of thermoneutral-housed ADGAT DKO mice. Relative mRNA levels of thermogenic genes in iWAT of chow-diet-fed male mice housed at thermoneutral temperature for 6 weeks (n=6). (**G**) Blood glucose levels were normal in thermoneutral-housed ADGAT DKO mice. Glucose levels in male mice fed a chow diet and housed at thermoneutral temperature for 6 weeks (n=8). Data are presented as mean ± SD. *p<0.05, **p<0.01, ***p<0.001.

The online version of this article includes the following figure supplement(s) for figure 5:

**Figure supplement 1.** Analysis of metabolites in inguinal white adipose tissue (iWAT) of ADGAT DKO mice.

**Figure supplement 2.** Primary adipocytes cultured from inguinal white adipose tissue (iWAT) of ADGAT DKO mice contain lipid droplets.

**Figure supplement 3.** Fibroblast growth factor 21 (FGF21) levels were increased in ADGAT DKO mice.

A surprising feature of this mouse model is the apparent metabolic health, despite the reduced capacity to store lipids in the adipose tissue. Typically, loss of WAT leads to a condition known as lipodystrophy (*Mann and Savage, 2019*; *Patni and Garg, 2015*). Lipodystrophy patients nearly always present with many metabolic derangements, including ectopic lipid accumulation in the liver (hepatic steatosis), hypertriglyceridemia, and insulin resistance or diabetes (*Lawrence, 1946*; *Misra and Garg, 2003*; *Reitman, 2002*). A characteristic feature of lipodystrophy is the decrease in levels of adipose tissue-derived hormones, such as leptin (*Lim et al., 2021*; *Friedman, 2019*), and many derangements of lipodystrophies are corrected by leptin therapy (*Oral et al., 2002*; *Shimomura et al., 1999*).

Remarkably, the lipodystrophic ADGAT DKO mice with marked reductions in fat storage were metabolically healthy and did not develop metabolic derangements, such as diabetes or hepatic steatosis, even when fed an HFD. The metabolic health of the ADGAT DKO mice likely is due to their intact ability to make adipose tissue and to maintain adipose tissue endocrine function. These findings are consistent with previous data showing that cultured adipocytes differentiate normally in the absence of TG storage (*Harris et al., 2011*). Adipose tissue of ADGAT DKO mice maintained the ability to synthesize and secrete adipose-derived hormones, such as leptin, which is crucially absent in typical lipodystrophy (*Shimomura et al., 1999*; *Savage, 2009*; *Garg, 2004*; *Wang et al., 2010*). Leptin levels often correlate with adiposity and TG stores (*Maffei et al., 1995*), but ADGAT DKO mice exhibit a dissociation of TG stores with leptin expression, thereby showing that these parameters appear not to be causally related. Instead, leptin levels may be better correlated with other adipose properties, such as the number of adipocytes, as reflected in the observed correlation of leptin with adipose tissue mass.

Our studies revealed that, in response to the compromised ability to store TG in WAT and BAT, mice activate pathways of energy expenditure, including the generation of beige adipocytes and activation of BAT. Aspects of this phenotype were present even at thermoneutral conditions. The mechanisms underlying the activation of energy expenditure and beiging in ADGAT DKO mice are presently unclear, but may involve SNS stimulation. The lack of fatty acids in adipocytes may result in other available fuels (fatty acids and glucose) being routed to the BAT and beige adipocytes to maintain body temperature. One possible mechanism for the beiging and increased energy expenditure is that the ADGAT DKO mice secrete a factor (or factors) that activates the SNS and beiging pathway. Although currently such a factor remains to be identified, this model is reminiscent of global DGAT1 knockout mice (*Smith et al., 2000*), which exhibit increased energy expenditure, enhanced glucose metabolism, protection from diet-induced obesity, and increased leptin sensitivity (*Chen et al., 2002*; *Chen et al., 2003*; *Chen et al., 2004*). For global DGAT1 knockout mice, fat transplant studies suggested the adipose tissue is the source of such factors (*Chen et al., 2003*).

Hormones that activate beiging, such as FGF21, are also testable candidate factors that may contribute to the ADGAT DKO phenotype. Many of the beneficial metabolic effects that we found in ADGAT DKO mice, including increased energy expenditure, increased glucose uptake by BAT, and torpor and browning, are found in murine models with increased levels of FGF21 (*Fisher et al., 2012*; *Owen et al., 2014*; *Inagaki et al., 2007*; *Kwon et al., 2015*). However, an endocrine effect of FGF21 seems unlikely in these mice due to similar levels of the hormone in plasma of HFD-fed mice. We note, however, that a paracrine effect is not excluded. Crossing ADGAT DKO mice with FGF21 knockout mice would further test FGF21's contribution to the metabolic phenotype of these mice. It would also be interesting to inhibit both DGAT enzymes at early or late time points in adipocyte differentiation and determine if endocrine or paracrine factors were altered, which might explain the systemic effect on metabolism.

In summary, ADGAT DKO mice represent an intriguing model in which a marked reduction in the ability to store TG in adipocytes triggers organismal pathways of energy dissipation. This suggests that exceeding the capacity to store energy in adipocytes is somehow sensed and triggers thermogenesis in adipose tissue. This phenotype likely requires an intact adipocyte endocrine system, which was found in ADGAT DKO mice but is often deficient in other models of lipodystrophy. The exact mechanism for how a TG storage defect triggers energy dissipation is currently unclear, but unraveling of this mechanism could lead to new strategies for treating or reducing obesity.

# Materials and methods

**Key resources table**

| Reagent type (species) or resource | Designation | Source or reference | Identifiers | Additional information |
|---|---|---|---|---|
| Antibody | Rabbit polyclonal anti-DGAT1 | In house (*Chitraju et al., 2019*) | | Request Farese and Walther lab for antibodies |
| Antibody | Rabbit polyclonal anti-DGAT2 | In house (*Chitraju et al., 2019*) | | Request Farese and Walther lab for antibodies |

*Continued on next page*

*Continued*

| Reagent type (species) or resource | Designation | Source or reference | Identifiers | Additional information |
|---|---|---|---|---|
| Antibody | Rabbit polyclonal anti-UCP1 | Abcam | Cat# ab10983 | WB (1:1000) |
| Antibody | Total OXPHOS Rodent WB Antibody Cocktail | Abcam | Cat# ab110413 | WB (1:1000) |
| Antibody | Rabbit monoclonal anti-GAPDH | Cell Signaling Technology | Cat# 5174S | WB (1:5000) |
| Antibody | Mouse monoclonal anti-α-Tubulin | Sigma-Aldrich | Cat# T9026 | WB (1:5000) |
| Commercial assay or kit | Power SYBR Green PCR Master Mix | Life Technologies | Cat# 4368706 | |
| Commercial assay or kit | iScript cDNA Synthesis Kit | Bio-Rad | Cat# 170-8891 | |
| Commercial assay or kit | RNeasy Mini Kit | QIAGEN | Cat# 74106 | |
| Commercial assay or kit | QIAzol Lysis Reagent | QIAGEN | Cat# 79306 | |
| Commercial assay or kit | QIAshredder | QIAGEN | Cat# 79656 | |
| Commercial assay or kit | RNase-Free DNase Set | QIAGEN | Cat# 79254 | |
| Commercial assay or kit | SuperSignal West Pico | Thermo Fisher Scientific | Cat# 34580 | |
| Commercial assay or kit | SuperSignal West Femto | Thermo Fisher Scientific | Cat# 34095 | |
| Commercial assay or kit | Infinity Triglycerides Reagent | Thermo Fisher | Cat# TR22421 | |
| Commercial assay or kit | Infinity Cholesterol Reagent | Thermo Fisher | Cat# TR13421 | |
| Commercial assay or kit | Free Glycerol Reagent | Sigma-Aldrich | Cat# F6428 | |
| Commercial assay or kit | HR Series NEFA-HR(2) Color Reagent A | FUJIFILM Medical Systems | Cat# 999-34691 | |
| Commercial assay or kit | HR Series NEFA-HR(2) Solvent A | FUJIFILM Medical Systems | Cat# 995-34791 | |
| Commercial assay or kit | HR Series NEFA-HR(2) Color Reagent B | FUJIFILM Medical Systems | Cat# 991-34891 | |
| Commercial assay or kit | HR Series NEFA-HR(2) Solvent B | FUJIFILM Medical Systems | Cat# 993-35191 | |
| Chemical compound, drug | 1,2-Dioleoyl-rac-glycerol | Sigma-Aldrich | Cat# D8394 | |
| Chemical compound, drug | Oleoyl coenzyme A lithium salt | Sigma-Aldrich | Cat# O1012 | |
| Chemical compound, drug | Oleoyl [14C] Coenzyme A | American Radiolabeled Chemicals, Inc | Cat# ARC 0527 | |
| Chemical compound, drug | Oleic acid [14C] | American Radiolabeled Chemicals, Inc | Cat# ARC 0297 | |
| Chemical compound, drug | Thin Layer Chromatography Plates | Analtech | Cat# P43911 | |
| Chemical compound, drug | Insulin | Lilly Corporation | Humulin R (U 100) | |
| Chemical compound, drug | CL 316,243 hydrate | Sigma-Aldrich | Cat# C5976 | |
| Chemical compound, drug | Glycogen from bovine liver | Sigma-Aldrich | Cat# G0885 | |
| Chemical compound, drug | Protease inhibitors | Roche | Cat# 11873580001 | |
| Other | ADGAT DKO mice | This paper | | Request Farese and Walther lab for mice |
| Other | Dgat1$^{flox/flox}$ mice | (*Shih et al., 2009*); The Jackson Laboratory | Cat# 017322 | |
| Other | Dgat2$^{flox/flox}$ mice | (*Chitraju et al., 2019*); The Jackson Laboratory | Cat# 033518 | |
| Other | Adipoq-Cre (Adiponectin-Cre) mice | (*Eguchi et al., 2011*); The Jackson Laboratory | Cat# 028020 | |

*Continued*

| Reagent type (species) or resource | Designation | Source or reference | Identifiers | Additional information |
|---|---|---|---|---|
| Other | Chow diet | PicoLab Rodent Diet 20 | Cat# 5053 | |
| Other | Breeder's diet | PicoLab Rodent Diet 20 | Cat# 5058 | |
| Other | High-fat diet | Envigo | Cat# TD.88137 | |

### Contact for reagent and resources sharing

Further information and requests for reagents and resources should be mailed to Robert V Farese, Jr. (rfarese@mskcc.org) and Tobias C Walther (TWalther@mskcc.org).

### Ethical approval of work

All mouse experiments were performed under the guidelines from Harvard Center for Comparative Medicine (protocol number: 05121).

### Generation of ADGAT DKO mice

To generate adipose tissue-specific *Dgat1* and *Dgat2* double-knockout (ADGAT DKO) mice, we crossed adipose tissue-specific *Dgat1* knockout mice (Cre-transgene expressed under control of the mouse adiponectin promoter [*Eguchi et al., 2011*] with *Dgat2* flox mice [*Chitraju et al., 2019*]) (Jackson Laboratory stock number: 033518). Two *Lox P* sites flank on either side of exons 14–17 in *Dgat1* gene; and exons 3–4 in *Dgat2* gene, respectively. To quantify *Dgat1* mRNA levels by qPCR, a forward primer residing on exon 14 and a reverse primer residing on exon 15 were designed. To quantify *Dgat2* mRNA levels by qPCR, a forward primer residing on exon 6 and a reverse primer residing on exon 7 were designed. Primer sequences are listed in *Supplementary file 1*.

### Animal husbandry

All mouse experiments were performed under the guidelines from Harvard Center for Comparative Medicine. Mice were maintained in a barrier facility, at room temperatures (22–23°C), on a regular 12 hr light and 12 hr dark cycle and had ad libitum access to food and water unless otherwise stated. For thermoneutral studies, mice were housed at 29°C. Mice were fed on standard laboratory chow diet (PicoLab Rodent Diet 20, 5053; less than 4.5% crude fat) or western-type HFD (Envigo, TD.88137; 21.2% fat by weight, 42% kcal from fat).

### Cold-exposure studies

For cold-exposure experiments (at 5°C), mice were single-housed in the morning around 8:00 am. Mice had free access to food and water unless otherwise stated. Core body temperatures were recorded using a rectal probe thermometer.

### DGAT activity assay

DGAT enzymatic activity was measured in WAT and BAT lysates at $V_{max}$ substrate concentrations. Assay mixture contained 20 µg of adipose tissue lysate, 100 µM of 1,2-dioleoyl-*sn*-glycerol, 25 µM of oleoyl-CoA, which contained [$^{14}C$] oleoyl-CoA as tracer, and 5 mM $MgCl_2$ in an assay in buffer containing 100 mM Tris-HCl (pH 7.4) and protease inhibitors. Reactions were carried out as described (*Cases et al., 2001*; *Chitraju et al., 2017*). After stopping the reaction, lipids were extracted and separated by TLC using a hexane:diethyl ether:acetic acid (80:20:1) solvent system. The TLC plates were exposed to phosphor imager screen and developed.

### Tissue lipid analysis

Approximately 50 mg of adipose tissue was homogenized in 1 ml of lysis buffer (250 mM sucrose, 50 mM Tris Cl, pH 7.0, with protease inhibitor cocktail [11873580001, Roche]). The homogenate was mixed with 5 ml of chloroform:methanol (3:2 vol:vol) and extracted for 2 hr by vigorous shaking. Upon centrifugation at 3000 × *g* at room temperature for 10 min, 100 µl of lower organic phase was collected and dried in a speed vac. To the dried lipids, 100–300 µl of 0.1% Triton X-100 was added, and the solution was sonicated using ultrasonic homogenizer (Biologics, Inc, model 3000MP) for 10 s with

30% amplitude. The total TG content was measured using the Infinity TGs reagent (Thermo Scientific) according to the manufacturer's protocol. TG and total cholesterol were measured using Infinity TGs reagent (Thermo Scientific) and a cholesterol E kit (Wako Diagnostics), respectively, according to the manufacturer's protocol. For plasma lipids measurement, 5 µl of plasma was used directly.

## Microscopy and image processing

Microscopy was performed on spinning disk confocal microscope (Yokogawa CSU-X1) set up on a Nikon Eclipse Ti inverted microscope with a 100× ApoTIRF 1.4 NA objective (Nikon) in line with 2× amplification. BODIPY 493/503 fluorophore was exited on 561 nm laser line. Fluorescence was detected by an iXon Ultra 897 EMCCD camera (Andor). Acquired images were processed using FIJI software (http://fiji.sc/Fiji).

## RNA extraction and quantitative real-time PCR

Total RNA from tissues was isolated with the Qiazol lysis reagent and using the protocol of the RNeasy Kit (QIAGEN). Complementary DNA was synthesized using the iScript cDNA Synthesis Kit (Bio-Rad), and qPCRs were performed using the SYBR Green PCR Master Mix Kit (Applied Biosystems).

## Immunoblotting

Tissues were lysed using RIPA lysis buffer (25 mM Tris Cl, pH 7.6, 150 mM NaCl, 1% NP-40, 1% sodium deoxycholate, 0.1% SDS) containing protease inhibitors (11873580001, Roche). Proteins were denatured in Laemmli buffer and separated on 10% SDS-PAGE gels and transferred to PVDF membranes (Bio-Rad). The membranes were blocked with blocking buffer for 1 hr in TBST containing 5% BSA or 5% milk, and then incubated with primary antibodies overnight. The membranes were then washed three times with TBST for 10 min, and incubated in mouse secondary antibodies (Santa Cruz Biotechnology) at 1:5000 dilutions in blocking buffer. Membranes was washed again three times with TBST for 10 min, and revealed using the Super Signal West Pico kit (Thermo Scientific).

## Comprehensive Lab Animal Monitoring System

Mice were housed individually and acclimatized for 2 days. Oxygen consumption, carbon dioxide release, energy expenditure, and activity were measured using a Columbus Instruments' Oxymax Comprehensive Lab Animal Monitoring System system according to guidelines for measuring energy metabolism in mice (*Tschöp et al., 2011*).

## Lipidomics analysis

For lipidomic analysis of iWAT, ~50 mg of iWAT was homogenized in 1 ml ice-cold phosphate-buffered saline using a bead mill homogenizer. Tissue lysates (50 µg) were transferred to a pyrex glass tubes with a PTFE-liner cap. Lipids were extracted by Folch method (*Folch et al., 1957*), briefly, 6 ml of ice-cold chloroform-methanol (2:1 vol/vol) and 1.5 ml of water were added to the samples, and tubes were vortexed thoroughly to mix the samples homogenously with a polar and non-polar solvent. SPLAH mix internal standards were spiked in before the extraction. The organic phase of each sample was normalized by total soluble protein amounts and measured by BCA assay (Thermo Scientific, 23225, Waltham, MA, USA). After vortexing, samples were centrifuged for 30 min at 1100 rpm at 4°C to separate the organic and inorganic phases. Using a sterile glass pipette, the lower organic phase was transferred into a new glass tube, taking care to avoid the intermediate layer of cellular debris and precipitated proteins. The samples were dried under nitrogen flow until the solvents were completely dried. Samples were resuspended in 250 µl of chloroform:methanol 2:1 and stored in –80°C until mass spectrometer (MS) analysis. Lipids were separated using ultra-high-performance liquid chromatography (UHPLC) coupled with tandem MS. Briefly, UHPLC analysis was performed on a C30 reverse-phase column (Thermo Acclaim C30, 2.1×250 mm, 3 µm operated at 55°C; Thermo Fisher Scientific) connected to a Dionex UltiMate 3000 HPLC system and a QExactive orbitrap MS (Thermo Fisher Scientific) equipped with a heated electrospray ionization probe. Five µl of each sample was analyzed separately, using positive and negative ionization modes. Mobile phase contained 60:40 water:acetonitrile (vol:vol), 10 mM ammonium formate, and 0.1% formic acid, and mobile phase B consisted of 90:10 2-propanol/acetonitrile, also including 10 mM ammonium formate and 0.1% formic acid. MS spectra of lipids were acquired in full-scan/data-dependent MS2 mode. For the full-scan acquisition,

the resolution was set to 70,000, the AGC target was 1e6, the maximum injection time was 50 ms, and the scan range was m/z=133.4–2000. For data-dependent MS2, the top 10 ions in each full scan were isolated with a 1.0 Da window, fragmented at a stepped normalized collision energy of 15, 25, and 35 units, and analyzed at a resolution of 17,500 with an AGC target of 2e5 and a maximum injection time of 100 ms. Peak identification and data analysis were carried out using Lipid Search software version 4.1 SP (Thermo Fisher Scientific) (*Taguchi and Ishikawa, 2010*).

## Metabolomic analysis

BAT and iWAT was snap-frozen in liquid nitrogen and ground at cryogenic temperature with a cyro-mill (Retsch, Newtown, PA, USA). The tissue was extracted with –20°C 40:40:20 methanol:acetonitrile:water at a concentration of 25 mg/ml. Samples were vigorously vortexed and centrifuged at 4°C at 16,000 × $g$ for 10 min, and the supernatant was transferred to LC-MS vials for analysis. Chromatographic separation was performed using XBridge BEH Amide XP Column (2.5 μm, 2.1 mm × 150 mm) with associated guard column (2.5 μm, 2.1 mm × 5 mm) (Waters, Milford, MA, USA). The mobile phase A was 95% water and 5% acetonitrile, containing 10 mM ammonium hydroxide and 10 mM ammonium acetate. The mobile phase B was 80% acetonitrile and 20% water, with 10 mM ammonium hydroxide and 10 mM ammonium acetate. The linear elution gradient was: 0–3 min, 100% B; 3.2–6.2 min, 90% B; 6.5–10.5 min, 80% B; 10.7–13.5 min, 70% B; 13.7–16 min, 45% B; and 16.5–22 min, 100% B. The flow rate was 0.3 ml/min. The autosampler was maintained at 4°C. The injection volume was 5 μl, and needle wash was performed between samples using 40:40:20 methanol:acetonitrile:water. The MS used was Q Exactive HF (Thermo Fisher Scientific, San Jose, CA, USA), and scanned from 70 to 1000 m/z with switching polarity. The resolution was 120,000. Metabolites were identified based on accurate mass and retention time using an in-house library, and the software used was El-Maven (Elucidata, Cambridge, MA, USA). Data was analyzed using R software (version 4.2.0). The ion intensity of each sample was first normalized to the corresponding sample protein content. Differentially abundant metabolites were analyzed with the limma R/Bioconductor package, and the multiple comparisons were corrected using the Benjamini-Hochberg procedure (adjusted p value; q value). The volcano plots were generated using the ggplot and ggrepel packages.

## Statistical analyses

Data are presented as mean ± SD (standard deviation). Statistical significance was evaluated by unpaired two-tailed Student's t-test or two-way ANOVA with Bonferroni's multiple comparison test. Significant differences are annotated as follows: *p<0.05, **p<0.01, ***p<0.001.

## Acknowledgements

We thank members of the Farese and Walther laboratory for helpful comments and G Howard for editorial assistance. We thank Karen Inouye and Sarah Mitchell for helping with indirect calorimetry analysis. Nathan Heinzman for helping with metabolomics experiment and the Longwood small animal imaging facility at Beth Israel Deaconess Medical Center for PET/CT analysis. This work was supported in part by NIH grant R01GM124348 (to RF). TCW is an investigator of the Howard Hughes Medical Institute.

## Additional information

### Funding

| Funder | Grant reference number | Author |
| --- | --- | --- |
| National Institutes of Health | R01GM124348 | Robert V Farese Jr |
| Howard Hughes Medical Institute | | Tobias C Walther |

The funders had no role in study design, data collection and interpretation, or the decision to submit the work for publication.

## Author contributions

Chandramohan Chitraju, Conceptualization, Data curation, Formal analysis, Validation, Investigation, Visualization, Methodology, Writing – original draft, Writing – review and editing; Alexander W Fischer, Investigation, Methodology, Writing – original draft, Writing – review and editing; Yohannes A Ambaw, Data curation, Formal analysis, Investigation, Methodology, Writing – original draft, Writing – review and editing; Kun Wang, Data curation, Formal analysis, Investigation, Visualization, Writing – original draft, Writing – review and editing; Bo Yuan, Formal analysis, Investigation, Visualization, Writing – original draft; Sheng Hui, Formal analysis, Investigation, Writing – original draft; Tobias C Walther, Conceptualization, Data curation, Supervision, Funding acquisition, Investigation, Writing – original draft, Project administration, Writing – review and editing; Robert V Farese Jr, Conceptualization, Data curation, Formal analysis, Supervision, Funding acquisition, Investigation, Visualization, Writing – original draft, Project administration, Writing – review and editing

## Author ORCIDs

Chandramohan Chitraju ![ORCID] https://orcid.org/0000-0001-8338-8976
Alexander W Fischer ![ORCID] https://orcid.org/0000-0001-6717-9090
Tobias C Walther ![ORCID] https://orcid.org/0000-0003-1442-1327
Robert V Farese Jr, ![ORCID] https://orcid.org/0000-0001-8103-2239

## Ethics

All mouse experiments were performed under the guidelines from Harvard Center for Comparative Medicine. Harvard IACUCProtocol number: 05121.

Reviewer #1 (Public Review): https://doi.org/10.7554/eLife.88049.3.sa1
Reviewer #2 (Public Review): https://doi.org/10.7554/eLife.88049.3.sa2
Reviewer #3 (Public Review): https://doi.org/10.7554/eLife.88049.3.sa3
Author Response https://doi.org/10.7554/eLife.88049.3.sa4

---

# Additional files

## Supplementary files
- MDAR checklist
- Supplementary file 1. List of primers used for quantitative real-time PCR analysis of mouse genes.
- Source data 1. Source data is provided as zip file.

## Data availability

All data generated or analysed during this study are included in the manuscript and supporting file.

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
