## [Editor Report · eLife assessment]

This study introduces a **valuable** paradigm in the field of adipose tissue biology: blocking triglyceride storage in adipose tissue does not lead to lipodystrophy and impaired glucose homeostasis but instead improves metabolic health. The evidence supporting these claims is **convincing**, based on a comprehensive metabolic analysis, although mechanistic studies would strengthen the study and its impact. This study will be of high interest to those in the adipose tissue biology and metabolism fields.

---

## [Referee Report · Reviewer #1 (Public Review)]

The present study examined the physiological mechanisms through which impaired TG storage capacity in adipose tissues affects systemic energy homeostasis in mice. To accomplish this, the authors deleted DGAT1 and DGAT2, crucial enzymes for TG synthesis, in an adipocyte-specific manner. The authors found that ADGAT DKO mice substantially lost the adipose tissues and developed hypothermia when fasted; however, surprisingly, ADGAT KO mice were metabolically healthy on a high-fat diet. The authors found that it was accompanied by elevated energy expenditure, enhanced glucose uptake by the BAT, and enhanced browning of white adipose tissues. This unique animal model provided exciting opportunities to identify new mechanisms to maintain systemic energy homeostasis even in a compromised energy storage capacity. Overall, the data are compelling and support the conclusions of the paper. The manuscript is also clearly written.

---

## [Referee Report · Reviewer #2 (Public Review)]

Here, Chitraju et al have studied the phenotype of mice with an adipocyte-specific deletion of the diglycerol acyltransferases DGAT1 and DGAT2, the two enzymes catalyzing the last step in triglyceride biosynthesis. These mice display reduced WAT TG stores but contrary to their expectations, the TG loss in WAT is not complete and the mice are resistant to a high-fat diet intervention and display a metabolically healthier profile compared to control littermates. The mechanisms underlying this are not entirely clear, but the double knockout (DKO) animals have increased EE and a lower RQ suggesting that enhanced FA oxidation and WAT "browning" may be involved. Moreover, both adiponectin and leptin are expressed in WAT and are detectable in circulation. The authors propose that "the capacity to store energy in adipocytes is somehow sensed and triggers thermogenesis in adipose tissue. This phenotype likely requires an intact adipocyte endocrine system...." Overall, I find this to be an interesting notion.

---

## [Referee Report · Reviewer #3 (Public Review)]

In this study, the authors sought to test the hypothesis that blocking triglyceride storage in adipose tissue by knockout of DGAT1 and DGAT2 in adipocytes would lead to ectopic lipid deposition, lipodystrophy, and impaired glucose homeostasis. Surprisingly, the authors found the opposite result, with DGAT1/2 DKO in adipocytes leading to increased energy expenditure, minimal ectopic lipid deposition, and improved glucose homeostasis with HFD feeding. These metabolic improvements were largely attributed to increased beiging of the white fat and increased brown adipose tissue activity. This study provides an interesting new paradigm whereby impairing fat storage, the major function of adipose tissue, does not lead to severe metabolic disease, but rather improves it. The authors provide a comprehensive assessment of the metabolism of these DKO mice under chow and HFD conditions, which support their claims. The study lacks in mechanistic insight, which would strengthen the study, but does not detract from the authors' major conclusions.

---

## [Author Response]

The following is the authors’ response to the original reviews.

**Reviewer #1 (Public Review):**
The present study examined the physiological mechanisms through which impaired TG storage capacity in adipose tissues affects systemic energy homeostasis in mice. To accomplish this, the authors deleted DGAT1 and DGAT2, crucial enzymes for TG synthesis, in an adipocyte-specific manner. The authors found that ADGAT DKO mice substantially lost the adipose tissues and developed hypothermia when fasted; however, surprisingly, ADGAT KO mice were metabolically healthy on a high-fat diet. The authors found that it was accompanied by elevated energy expenditure, enhanced glucose uptake by the BAT, and enhanced browning of white adipose tissues. This unique animal model provided exciting opportunities to identify new mechanisms to maintain systemic energy homeostasis even in a compromised energy storage capacity. Overall, the data are compelling and well support the conclusion of this paper. The manuscript is clearly written.

We thank the reviewer for the time invested to critically review our paper.

**Reviewer #2 (Public Review):**
Here, Chitraju et al have studied the phenotype of mice with an adipocyte-specific deletion of the diglycerol acyltransferases DGAT1 and DGAT2, the two enzymes catalyzing the last step in triglyceride biosynthesis. These mice display reduced WAT TG stores but contrary to their expectations, the TG loss in WAT is not complete and the mice are resistant to a high-fat diet intervention and display a metabolically healthier profile compared to control littermates. The mechanisms underlying this are not entirely clear, but the double knockout (DKO) animals have increased EE and a lower RQ suggesting that enhanced FA oxidation and WAT "browning" may be involved. Moreover, both adiponectin and leptin are expressed in WAT and are detectable in circulation. The authors propose that "the capacity to store energy in adipocytes is somehow sensed and triggers thermogenesis in adipose tissue. This phenotype likely requires an intact adipocyte endocrine system...." Overall, I find this to be an interesting notion.

We thank the reviewer for the time invested to critically review our paper.

**Reviewer #3 (Public Review):**
In this study, the authors sought to test the hypothesis that blocking triglyceride storage in adipose tissue by knockout of DGAT1 and DGAT2 in adipocytes would lead to ectopic lipid deposition, lipodystrophy, and impaired glucose homeostasis. Surprisingly, the authors found the opposite result, with DGAT1/2 DKO in adipocytes leading to increased energy expenditure, minimal ectopic lipid deposition, and improved glucose homeostasis with HFD feeding. These metabolic improvements were largely attributed to increased beiging of the white fat and increased brown adipose tissue activity. This study provides an interesting new paradigm whereby impairing fat storage, the major function of adipose tissue, does not lead to severe metabolic disease, but rather improves it. The authors provide a comprehensive assessment of the metabolism of these DKO mice under chow and HFD conditions, which support their claims. The study lacks in mechanistic insight, which would strengthen the study, but does not detract from the authors' major conclusions.

We thank the reviewer for the time invested to critically review our paper.

The conclusions of this paper are mostly well-supported, but some aspects should be clarified and extended.1. The authors claim the beiging of WAT of ADGAT DKO mice is partially through the SNS; however, housing these mice at thermoneutrality did not block the beiging, which seems to negate that claim. Is there evidence of increased cAMP/PKA activation in the adipose tissues of ADGAT DKO to support the premise that the beiging is activated by the SNS, even at thermoneutrality? Alternatively, if the authors block beta-adrenergic receptors with antagonists, such as propranolol, does this block the beiging?

We are currently unsure of the mechanism(s) for WAT beiging and whether it requires the SNS. We attempted denervation experiments to ablate SNS input; however, the results were consistent with partial denervation and not clearly interpretable, so we elected not to include them in the manuscript. Unfortunately, we did not measure cAMP/PKA activation or utilize beta blockers in attempt to block SNS activation. Due to a recent laboratory move, there are no study mice available to perform these experiments.

2. It's been shown that autocrine FGF21 signaling is sufficient to promote beiging of iWAT (PMID 34192547). The authors show Fgf21 mRNA is increased in iWAT of chow-fed ADGAT DKO mice. Is Fgf21 also increased in iWAT of HFD-fed mice? This and measurement of local FGF21 secretion by adipocytes would strengthen this study.

We thank the reviewer for this question. Unfortunately, we did not measure Fgf21 mRNA levels in iWAT of HFD-fed mice or FGF21 secretion by adipocytes and mice are not currently available. We agree, however, that FGF21 is a candidate for mediating this phenotype. Testing this idea would likely require crossing the ADGAT DKO mice with FGF21 KO mice. Arguing against FGF21 as contributing systemically, plasma levels were similar in HF-fed ADGAT DKO mice and controls.

3. The primary adipocytes in Figure 5–figure supplement 2A do not appear to have any depletion in TG stores, suggesting this may not be an appropriate model to study the cell autonomous effects of ADGAT DKO on beiging. The authors should use DGAT inhibitors instead to corroborate or investigate this question.

We agree with the reviewer that primary adipocytes from ADGAT DKO mice may not be the best model to study the cell autonomous effects of beiging, particularly since they are accumulating lipids. On the other hand, it’s not likely that DGAT inhibitors would be any better than the genetic deletions of the enzymes. Presumably, the neutral lipids are being synthesized by enzymes other than DGAT1 or DGAT2.

4. Multiple studies have shown the importance of lipolysis for the activation of brown and beige thermogenic programs (PMID 35803907, 34048700) and can be potentiated by HFD feeding (PMID 34048700). In the absence of DGAT activity in ADGAT DKO mice, it seems plausible that free fatty acids could be elevated, especially in the context of HFD. Are free fatty acids elevated in the adipose tissues, which could promote thermogenic gene expression?

We thank the reviewer for pointing this out. Although we cannot exclude this mechanism, arguing against it, we found lower levels of almost all free fatty acid species in iWAT of chow diet fed ADGAT DKO (Figure 5–figure supplement 1, metabolomics). Additionally, plasma FFA were reduced in these mice.

5. The lack of ectopic lipid deposition in the ADGAT DKO mice is striking, especially under HFD conditions. Can the increased energy expenditure fully account for the difference in whole body fat accumulation between Control and DKO mice or have the mice activated other energy disposal mechanisms? Please discuss or include measurement of fat excretion in the feces to strengthen this study.

Although decreased lipid absorption may conceivably contribute to energy loss, we would not expect this to occur in adipocyte-specific knockout mice, and we did not measure the lipid content in the feces. We have added a discussion point to the manuscript.

**Reviewer #1 (Recommendations for the Authors):**
The authors wish to clarify the following points to strengthen this exciting work further.1. The authors demonstrated that DKO mice exhibited enhanced browning of WAT even under a thermoneutral condition, and this occurred in a non-cell autonomous fashion. Accordingly, the authors suggested the possibility that SNS activity was enhanced in DKO mice. It would be intriguing to examine the extent to which lipolysis is indeed enhanced in DKO mice. For instance, do DKO mice have higher FFA and glycerol levels than controls in circulation? This could explain a part of the phenotype, as a recent work suggested that WAT lipolysis triggers beige progenitor cell proliferation in WAT.

We thank the reviewer for this question. Although this is an interesting idea, we found that ADGAT DKO mice have lower levels of free fatty acids and glycerol in the circulation, indicating lipolysis not likely the underlying mechanism for increased beiging in ADGAT DKO mice. FFA were also reduced in the iWAT of the ADGAT DKO mice (as shown in supplemental data for Figure 5).

2. The authors suggested the possibility that other candidates in the DGAT2 gene family might compensate for the lack of DGAT1 and DGAT2. It will be insightful if the authors elaborate on this part - e.g., discussing any transcriptional changes of DGAT2 family members in the WAT of DKO mice.

We thank the reviewer for this question. Previous studies showed (Yen et al., 2005), monoacylglycerol acyl transferase (MGAT) enzymes also possess some TG synthesis activity. We found increased mRNA expression of MGAT1 and MGAT2 enzymes in white adipose tissue of ADGAT DKO mice. We now included this data in Figure 1–figure supplement 1G.

3. Minor: Statistics of the AUC for the GTT and ITT (Fig. 3G and 3H).

We thank the reviewer for this point. We have now updated the figures with statistics of the AUC for GTT and ITT.

**Reviewer #2 (Recommendations for the Authors):**
1. The authors suggest that the DKOs are protected against a high-fat diet due to an intact endocrine function combined with increased FA oxidation and WAT browning. This phenotype is interesting but as the authors write, the underlying mechanisms remain unclear. Furthermore, how important is retained endocrine function, in relation to WAT browning, in explaining the resistance to a high-fat diet in the DKOs? As these mice are born with the double DGAT KO and it is possible that compensatory mechanisms explain some of the observed effects. What happens with the endocrine function/browning effect if DGAT1/2 is inhibited in cells that already contain full TG stores? While I understand that studies in an inducible KO model are outside the scope of this study, data in cells where the effects of DGAT inhibition are studied early and late during differentiation would be interesting and could at least be discussed.

We thank the reviewer for this interesting question. It is possible that ADGAT DKO have adipose tissue-derived factors that act in endocrine or paracrine manner to induce beiging. Unfortunately, we do not have data addressing this point, but added discussion of this point to the revised manuscript.

2. While the possibility to dissociate TG storage from the endocrine function of WAT is attractive, the authors have only studied two adipokines. Do they have data on any other adipokine(-s) supporting the claim that the secretory function is intact?

We thank the reviewer for this question. Regrettably we did not measure additional adipokines, and the mice are no longer available for study due to a recent lab move.

**Reviewer #3 (Recommendations for the Authors):**
Minor comments/suggestions:1. The authors show multiple phospholipid species were increased by ADGAT DKO. Cardiolipin has been shown to promote brown fat thermogenesis (PMID 29861389). Were cardiolipin levels changed by ADGAT DKO?

We thank the reviewer for this question. We found cardiolipin levels were increased in iWAT of ADGAT DKO mice. However, we have not measured cardiolipin levels in brown fat.

2. A recent study (PMID 36914626) has shown that inhibition of lipogenesis in adipose tissue impairs autophagy and also causes beiging of white adipose tissue. Is autophagy affected by ADGAT DKO? Are de novo lipogenesis enzymes affected by the DKO?

We thank the reviewer for this interesting suggestion. We did find that mRNA levels of genes involved in de novo lipogenesis (Srebp1c, Acc, Fas) were decreased in iWAT of ADGAT DKO mice, as expected from some of our other studies involving DGAT inactivation. Unfortunately, we did not measured autophagy per se in iWAT of ADGAT DKO mice.